# Residential Nitrogen Dioxide Exposure and Cause-Specific Cerebrovascular Mortality: An Individual-Level, Case-Crossover Study

**DOI:** 10.3390/toxics12010010

**Published:** 2023-12-22

**Authors:** Yifeng Qian, Renzhi Cai, Xiaozhen Su, Qi Li, Shan Jin, Wentao Shi, Renjie Chen, Chunfang Wang, Jia He

**Affiliations:** 1Department of Health Statistics, Naval Medical University, Shanghai 200433, China; 2Department of Oral and Craniomaxillofacial Surgery, Shanghai Ninth People’s Hospital, College of Stomatology, Shanghai JiaoTong University School of Medicine, Shanghai 200011, China; 3Division of Vital Statistics, Institute of Health Information, Shanghai Municipal Center for Disease Control and Prevention, Shanghai 200336, China; 4School of Public Health, Key Lab of Public Health Safety of the Ministry of Education and NHC Key Lab of Health Technology Assessment, Fudan University, Shanghai 200032, China; 5Clinical Research Unit, School of Medicine, Shanghai Ninth People’s Hospital, Shanghai Jiao Tong University, Shanghai 200011, China

**Keywords:** cerebrovascular diseases, ischemic stroke, hemorrhagic stroke, sequelae of cerebrovascular disease, nitrogen dioxide, air pollution

## Abstract

Background: Existing studies have already shown a connection between nitrogen dioxide (NO_2_) exposure and cerebrovascular mortality. However, the differential effects of NO_2_ on cerebrovascular disease and its subtypes remain unclear and require further exploration. Methods: Daily stroke mortality data between 2013 and 2021 in Shanghai, China were collected. Residential daily air pollution data for each decedent were predicted from a satellite model. An individual-level, time-stratified, case-crossover design was applied to examine the relationship between NO_2_ exposure and cerebrovascular mortality. A combination of conditional logistic regression and distributed lag models with a maximum lag of 7 days was used for data analysis. Results: A total of 219,147 cases of cerebrovascular mortality were recorded. Among them, the proportion of sequelae of cerebrovascular disease, hemorrhagic stroke and ischemic stroke was 50.7%, 17.1% and 27.5%, respectively. The monotonic increases in mortality risks of cerebrovascular diseases, sequelae of cerebrovascular disease and ischemic stroke were observed, without any discernible thresholds. Each 10 μg/m^3^ increase in NO_2_ concentration was associated with increments of 3.62% [95% confidence interval (CI): 2.56%, 4.69%] for total cerebrovascular mortality, 4.29% (95% CI: 2.81%, 5.80%) for sequelae of cerebrovascular disease mortality and 4.30% (95% CI: 2.30%, 6.33%) for ischemic stroke mortality. No positive associations between NO_2_ exposure and hemorrhagic stroke mortality were observed. A greater risk of NO_2_ was observed in the warm season, in patients with less than 9 years of education and in those with single marital status. The effects of NO_2_ were robust to mutual adjustment of co-pollutants. Conclusions: Short-term exposures to NO_2_ may increase the risk of cerebrovascular mortality, specifically for ischemic stroke and sequelae of cerebrovascular disease.

## 1. Introduction

Cerebrovascular disease encompasses a range of conditions that impact the blood vessels that provide nourishment to the brain [1,2]. As a most common classification of cerebrovascular diseases, stroke has been the second major cause of death globally, with an estimate of 6.5 million attributable deaths worldwide and 2.2 million in China in 2019 [3]. Ischemic stroke and hemorrhagic stroke are the primary forms of cerebrovascular disease. Ischemic stroke is triggered by a blockage or constriction of blood vessels within the brain, while hemorrhagic stroke is the result of bleeding occurring inside or around the brain. Both types of cerebrovascular disease can result in neurological deficits, including paralysis, coordination difficulties and cognitive impairments [4,5]. Recognizing the factors that contribute to cerebrovascular disease is crucial for minimizing the incidence rate and to alleviate the burden of this condition.

Nitrogen dioxide (NO_2_) is a major component of air pollution produced primarily by combustion processes, such as motor vehicles, power plants and industrial facilities [6]. There has been an increasing number of epidemiological studies exploring the potential association between NO_2_ exposure and the incidence, hospitalization and mortality of cerebrovascular disease. However, most studies have utilized a time-series study design at the population level, which makes it difficult to control for individual confounding factors. Moreover, associations have been observed using data from environmental monitor stations, which introduces significant measurement errors. In addition, evidence was inconsistent on the association between NO_2_ exposure and cerebrovascular disease. For example, a meta-analysis involving 18 countries and over 230,000 participants suggested that short-term NO_2_ exposure may increase the incidence and mortality risk of stroke, with a 10 μg/m^3^ increase in NO_2_ concentration corresponding to an increased relative risk of 0.9% [95% confidence interval (CI): 0.3%, 1.6%] [7]. Another nationwide time-series study observed a significant correlation between a rise in NO_2_ concentration and increased hospitalizations due to ischemic stroke [relative risk (RR): 1.82% (95% CI: 1.45%, 2.19%)] [8]. On the other hand, negative findings from studies conducted in China and Ireland have revealed no substantial link between exposure to NO_2_ pollution and hospitalization for stroke [9,10].

The varying impacts of NO_2_ exposure on the development of various types of cerebrovascular diseases have been compared in some studies, yielding inconsistent results. A meta-analysis involving 28 countries reported that short-term exposure to NO_2_ was positively associated with the risk of mortality due to stroke. Each 10 μg/m^3^ increase in NO_2_ concentration was associated with increments of 1.4% (95% CI: 0.9%, 1.9%) for total stroke mortality, specifically 2.4% (95% CI: 1.0%, 3.8%) for ischemic stroke and 2.4% (95% CI: 0.3%, 4.5%) for hemorrhagic stroke [11]. In contrast, a separate case-crossover study conducted in China discovered that NO_2_ exposure was only linked to an increased risk of total stroke and ischemic stroke mortality, but not hemorrhagic stroke mortality [12]. Additionally, compared to ischemic and hemorrhagic stroke, there is a scarcity of research examining the association between NO_2_ exposure and the sequelae of cerebrovascular disease.

Therefore, we conducted an individual-level, time-stratified, case-crossover design, with the objective to quantitatively estimate the effects of short-term residential NO_2_ exposure on mortality due to different subtypes of cerebrovascular diseases in Shanghai, the largest city of China.

## 2. Materials and Methods

We collected mortality data at the individual level from the database of Shanghai Municipal Center for Disease Control and Prevention, spanning the period from 1 January 2013 to 31 December 2021. In Shanghai, death certificates are completed by either community doctors for deaths occurring outside of hospitals or hospital doctors for deaths within hospitals. In cases within hospitals, causes of death are determined through a combination of symptoms and CT or MRI examinations, following standard medical norms. For cases occurring outside the hospital, community doctors diagnose based on medical records and symptoms. The information on these certificates was coded according to the International Classification of Diseases, Revision 10 (WHO 1993). Deaths with codes I60-I69 were identified as cerebrovascular disease, specifically I60-I62 as hemorrhagic stroke, I63 as ischemic stroke and I69 as sequelae of cerebrovascular disease. Other types of cerebrovascular diseases (I64-I68) were not considered due to the small quantity of deaths. In addition, the death records included details such as sex, age, education, marital status, residential address and death date. Individuals without basic information were excluded from this study.

Through the integration of aerosol optical depth and meteorological data, a gap-filling approach utilizing random forest algorithms has been formulated for the precise prediction of ground-level air pollutants at a detailed spatiotemporal scale, which has high accuracy with the overall cross-validation R^2^ exceeding 0.80 compared to monitoring data [13,14,15]. In the current study, we estimated NO_2_, PM_2.5_, PM_10_ and O_3_ concentrations at the individual level by assigning the gridded concentration predictions to the addresses of deceased individuals, and calculated daily concentrations of PM_2.5–10_ by subtracting the concentrations of PM_2.5_ and PM_10_, with a spatial resolution of 1 km. In addition, daily temperature and humidity data were obtained from the fifth generation of European Reanalysis database, with spatial resolutions of 10 and 25 km, respectively [16].

The time-stratified case-crossover approach has been widely used to investigate the association between short-term exposure to air pollutants and health outcome as a self-controlled study. In our analyses, we selected control days to match the same day of the week in the same month and year when a death occurred. By comparing the exposure distribution during the case period with ones during control periods, this design effectively controls for confounding factors that remain relatively stable over a short period, such as sex, age, socioeconomic status and education level. The distributed lag linear model (DLM) and the distributed lag nonlinear model (DLNM) are statistical modeling techniques to estimate the connection between changing exposure and the health outcomes. By considering both lagged effects and non-linear associations, DLNM offers a more comprehensive understanding of the intricate relationship between exposure and outcomes as it evolves over time.

A combination of conditional logistic regression and distributed lag models with a maximum lag of 7 days was conducted to examine the cumulative associations between daily ambient NO_2_ exposure and cerebrovascular mortality. The single-pollutant models were constructed firstly, wherein DLM was used to assess the lag pattern and DLNM was employed to explore the exposure–response relationships, with a natural cubic spline function of 3 degrees of freedom (*df*) in a lag space of “cross-basis”. Additionally, daily temperature at the same lag of air pollutants and relative humidity at the concurrent day were introduced into the model for further adjustment, using 6 and 3 *df*, respectively. A binary indicator of public holidays was also included in our model.

We conducted stratified analyses to examine the potential modifying effects of sex, age, season, education levels and marital status. The warm season was defined as the period from 1 April to 30 September, while the cold season was from 1 October to 28 February. The status of unmarried, divorced and widowed were amalgamated into the “single” status category. Effect estimates within each stratum were compared using a two-sample *z* test to test the statistical significance of differences, and the corresponding 95% CI were calculated. The theory of this method has been previously detailed and references to the relevant formulae were made [17,18]. In addition, we further performed a sensitivity analysis to assess the robustness of the results, considering the potential impact of other air pollutants at a lag of 7 days using both a two-pollutant model and a multi-pollutant model.

We reported effect estimates as the percentage change in RRs of cerebrovascular mortality associated with a 10 μg/m^3^ increase of NO_2_ exposure. The reported *p* values were derived from two-sided tests at a significance level of α = 0.05. All statistical analyses were performed using R, V.4.2.1 (R Development Core Team 2010, https://www.scirp.org/reference/referencespapers?referenceid=493779, accessed on 21 October 2023).

The Institutional Review Board at the School of Public Health, Fudan University approved the study protocol (IRB#2021−04-0889) and waived the requirement for informed consent since the data were collected for administrative purposes and analyzed without any personal identifiers.

## 3. Results

A total of 219,147 cases of cerebrovascular mortality were recorded. Among them, there were 111,182 cases of sequelae of cerebrovascular disease, 37,539 cases of hemorrhagic stroke and 60,217 cases of ischemic stroke, accounting for 50.7%, 17.1% and 27.5%, respectively. Out of the total cases, 50.3% were female, 69.2% were aged 80 years or older, 56.0% occurred during the cold season, 41.6% had received at least 9 years of education and 53.1% were married (Table 1).

The summary statistics for air pollution concentrations and meteorological variables were shown in Table 2. The daily average concentrations for NO_2_, PM_2.5_, PM_2.5–10_ and O_3_ were 23.8 μg/m^3^, 43.1 μg/m^3^, 27.2 μg/m^3^ and 83.3 μg/m^3^, respectively. The daily temperature exhibited an average of 15.9 °C, while the relative humidity averaged at 72.9%. NO_2_ showed positive correlations with PM_2.5_ and PM_2.5–10_, but negative correlations with O_3_ and meteorological factors according to Spearman analysis (Table 3).

The lag structures for percent changes of the RR of cerebrovascular mortality associated with a 10 μg/m^3^ increase in NO_2_ were presented in Figure 1. The RR for mortality from cerebrovascular disease, sequelae of cerebrovascular disease and ischemic stroke reached its highest value on lag0, followed by a monotonic decrease, becoming statistically insignificant after 6 days (Appendix A). No positive associations between NO_2_ exposure and hemorrhagic stroke mortality were observed at any lag periods.

Figure 2 depicted the cumulative exposure–response curves illustrating the connection between NO_2_ and cerebrovascular mortality over a period of 0 to 7 days. The findings indicated a consistent rise in mortality risks for all types of cerebrovascular diseases, except for hemorrhagic stroke. There was no identifiable threshold in these relationships, and the curves exhibited a nearly straight-line pattern. Each 10 μg/m^3^ increase in NO_2_ concentration was associated with increments of 3.62% (95% CI: 2.56%, 4.69%) for total cerebrovascular disease, 4.29% (95% CI: 2.81%, 5.80%) for sequelae of cerebrovascular disease and 4.30% (95% CI: 2.30%, 6.33%) for ischemic stroke (Table 4). Additionally, the stratification analysis revealed that NO_2_ was positively associated with all subgroups of total cerebrovascular and sequelae of cerebrovascular mortality. The adverse effect of NO_2_ was substantially higher in the warm season in terms of sequelae of cerebrovascular mortality. Patients with more than 9 years of education had a significantly lower risk of total cerebrovascular mortality. For patients in single status, the estimated risks of total cerebrovascular and ischemic stroke mortality were significantly higher compared to their counterpart (Appendix A).

In the sensitivity analysis, we examined the robustness of our results using a two-pollutant model to control for PM_2.5_, PM_2.5–10_ and O_3_, respectively. In addition, we further utilized multi-pollutant models adjusting for all other air pollutants, including PM_2.5_, PM_2.5–10_ and O_3_. As shown in Table 5, the effect estimates of NO_2_ on various cerebrovascular disease mortality were robust and varied slightly after controlling for the effects of PM_2.5_, PM_2.5–10_ and O_3_.

## 4. Discussion

In an individual-level time-stratified case-crossover design, our study summarized the relationship between residential NO_2_ exposure and cerebrovascular mortality over a nine-year period in Shanghai, China. Our finding indicated that a temporary increase in NO_2_ concentration corresponded to an increased risk of mortality from total cerebrovascular diseases, sequelae of cerebrovascular disease and ischemic stroke. Notably, an elevated risk of NO_2_ exposure was observed, especially during the warm season, in patients with lower levels of education and those with single marital status. In contrary, no significant evidence of effect modification by age and sex was found.

Consistent with previous findings from time-series and case-crossover studies, we analyzed the acute health effects of ambient NO_2_ pollution, aiming to explore the risk of premature death caused by the acute exacerbation of air NO_2_ pollution in cerebrovascular patients. Our results indicated that a rise in NO_2_ concentration was significantly associated with an increased risk of cerebrovascular mortality, consistent with previous research. For instance, a time-series study based on seven-year death records in central China showed that, for every 10 μg/m^3^ increase in NO_2_ concentration, the risk of cerebrovascular disease mortality increased [ERR: 1.090 (95% CI: 0.822, 1.358) [19]. Another time-series study conducted by Dong et al. in China also revealed that short-term exposure to NO_2_ could significantly increase the risk of ischemic stroke, with a 0.208% (95% CI: 0.036%, 0.381%) increase in hospital admissions and a 0.263% (95% CI: 0.004%, 0.522%) increase in mortality [20]. A time-series study in a large metropolitan area discovered a significant connection between ambient NO_2_ exposure and acute cerebrovascular events [21]. There are several research studies suggesting a substantial association between ambient air NO_2_ pollution and ischemic stroke and hemorrhagic stroke. For example, a time-stratified case-crossover study among 412,567 stroke deaths in Jiangsu, China reported that each 10 μg/m^3^ increase of NO_2_ concentration was linked to a 2.90% increase in odds of mortality from ischemic stroke, significantly stronger than hemorrhagic stroke of 1.15% [22]. Additionally, negative associations between NO_2_ exposure and hemorrhagic or ischemic stroke have been verified by several other studies. A time-stratified case-crossover study conducted in the Beibu Gulf Region of China reported that an elevated NO_2_ exposure of 11.2 μg/m^3^ was significantly linked to an increased risk for hospitalizations of total stroke [OR: 1.040, 95% CI: 1.027–1.053] and ischemic stroke [OR: 1.052, 95% CI: 1.033–1.071], but there was no significant association with hemorrhagic stroke [12]. A multicenter case-crossover study also suggested an evident relationship between hourly exposure to NO_2_ and an increased risk of hospital admissions for total stroke and ischemic stroke, while no evident impact was found on hemorrhagic stroke [23]. Another time-stratified case-crossover study among participants of the Women’s Health Initiative observed a positive association between risk for incidence of hemorrhagic stroke and NO_2_ exposure, but negative association of total stroke, ischemic stroke or ischemic stroke subtypes [24]. The time-series analyses based on the new rural cooperative medical scheme in East China found that a 10 μg/m^3^ increase of NO_2_ was related to an elevated risk of ischemic stroke (RR: 1.021, 95% CI: 1.006–1.035), while there was no significant relationship to hemorrhagic stroke [25]. There was considerable variation in the results of previous studies.

The variations in the effects of NO_2_ exposure on the risk of cerebrovascular mortality may be attributed to several factors including differences in study design, model parameters, exposure measurement, local geographical and climatic characteristics, source composition and the quality of health data. Firstly, the individual-level, time-stratified, case-crossover study design that we adopted can automatically control for individual confounding factors that do not change over a short period of time, addressing a common challenge in time-series studies. Secondly, the choice of model parameters can also substantially influence research results. We utilized distributed lag models to assess the cumulative effects during lag periods, different from commonly used single-day or moving average lag approaches in previous research. Thirdly, the method of exposure assessment is crucial to the results. It is particularly important to emphasize that we obtained residential NO_2_ concentration with a higher resolution exposure model, allowing for a more precise estimation of its impact on the risk of cerebrovascular mortality, particularly in suburban and rural regions. Fourthly, inconsistencies in research findings may also arise from variations in local weather and source composition. Areas near combustion sites often exhibit a heightened concentration of pollutants, with ultrafine particles potentially playing a significant role. However, due to limited data availability, the impact of ultrafine particles is not well comprehended. Particle mass is an inadequate proxy for ultrafine particles and their number. Depending on the sources and local geography and meteorology, NO_2_ can serve as a very good indicator of exposure to ultrafine particles. Therefore, some effects attributed to NO_2_ might actually be caused by other combustion byproducts. Finally, inconsistencies could result from differences in quality of health data, especially regarding the type of stroke (ischemic or hemorrhagic). For deaths that occurred outside of hospitals, the determination of the cause of death is often based solely on symptoms. However, distinguishing between ischemic and hemorrhagic stroke based solely on symptoms can be challenging. In the current study, 65.28% of the mortality cases occurred in hospitals, while 33.46% occurred outside of hospitals. Among the hospital cases, the cause of death was established through examinations such as CT or MRI, while in the other cases, 95.55% were diagnosed based on medical records. Therefore, we anticipate a minimal misclassification rate in our study.

In the current study, we further found that NO_2_ significantly increased the risk of mortality from sequelae of cerebrovascular disease, which has been rarely investigated in previous studies. For example, Wang et al. reported no substantial association between short-term NO_2_ exposure and hospital admission for sequelae of stroke in Chinese elderly [26]. The inconsistency observed may be related to the difference of exposure levels, population susceptibility and sociodemographic characteristics. It is important to emphasize that the diagnostic outcomes for sequelae of cerebrovascular diseases are not clearly defined. Mortality from this condition may be linked to complications induced by NO_2_, such as respiratory infections and thrombotic diseases, or from new cerebrovascular events that are difficult to categorize and necessitate a more precise diagnosis. Nonetheless, our findings provide new epidemiological evidence regarding the effect of NO_2_ on the mortality from sequelae of cerebrovascular diseases, and further research in this area is warranted.

The stratified analysis showed that patients who are single have lower levels of education, were exposed to air NO_2_ pollution during the warm season and had a higher relative risk cerebrovascular mortality. These epidemiological findings can provide important insights for toxicology research. Compared to single individuals, married people are more likely to receive comprehensive care and have an improved quality of life. This may, in turn, reduce their susceptibility to cerebrovascular mortality. Individuals with lower educational backgrounds often experience inferior living and working conditions, are more prone to poor health status and possess limited knowledge about disease control and prevention. As a result, they may have heightened sensitivity to environmental air pollution. In addition, during the warm season, outdoor activities and increased ventilation through open windows are more common than in cold season, resulting in reduced measurement error and a more accurate assessment of the risk associated with NO_2_. In contrast, the risks during the cold season may be underestimated [10].

The precise mechanisms underlying the association between NO_2_ pollution exposure and cerebrovascular disease remain incompletely understood, but several potential pathways have been proposed. One significant triggering mechanism involves the activation of autonomic respiratory reflex arcs by pollutants, which occurs through pulmonary receptors, baroreceptors and chemical receptors. This activation can result in increased vascular resistance and the potential development of arrhythmias and hypertension [27]. Following inhalation, gaseous pollutants can enter the bloodstream and react with nitric oxide to form oxygen species. As a result, the rapid interaction can lead to endothelial dysfunction within the systemic circulation [28]. Moreover, NO_2_ is the representative pollutant of vehicle exhaust gas. Researchers have reported that NO_2_ exposure can lead to airway and systemic inflammations, promote impaired endothelial function and increase blood viscosity, which contributed to vascular damage and thrombosis, ultimately leading to cerebrovascular diseases. A double-blind randomized crossover study suggested that exposure to diesel exhaust in individuals without existing health conditions can enhance the activation of platelets and the formation of blood clots at injured blood vessels [29]. These mechanisms are more closely related to the pathophysiology of ischemic stroke, which may be the reason why ischemic stroke is more susceptible to NO_2_ pollution compared with hemorrhagic stroke. In addition, individuals with sequelae of cerebrovascular disease often experience poorer physical conditions, including motor, sensory and cognitive impairments, which can lead to reduced immunity and increased sensitivity to NO_2_ air pollution.

There are several advantages of our research. Firstly, we obtained residential daily air pollution data from a satellite model with a resolution of 1 km, thereby improving the precision of our statistical analysis. Secondly, we differentiated the impact of NO_2_ on various subtypes of cerebrovascular diseases. Finally, we conducted an individual-level, time-stratified, case-crossover design, which reduces residual confounding compared to a time-series design. However, the current study also has certain limitations. We only examine the impacts of outdoor NO_2_ exposure, without considering indoor NO_2_ levels, and the study was conducted only in a single city. In addition, the diagnosis of cerebrovascular subtypes may have some errors for out-of-hospital mortality by symptoms and medical records, although this error is considered unrelated to air pollution and it is not a confounding factor. Furthermore, although our satellite data have the strength of effectively capturing the spatial variability of NO_2_ better than the city-wide monitoring data, there is room for improvement in estimating of the effect of NO_2_ on the risk of cerebrovascular mortality, as we do not have exact information on the time each person spent at his or her home address. Nevertheless, nearly 70% of participants in our study were aged over 80, with an average age of 81.9. We believe that these people are more likely to spend their time near their home address, given that a super majority of them have retired and may be physically challenged at the same time. Finally, there remains a possibility of residual biases or unobserved factors influencing our results, despite the use of a time-stratified case-crossover design to control for confounding variables that do not change over time.

## 5. Conclusions

In summary, the findings of the present study suggested that residential NO_2_ exposure consistently increased the risks of mortality of cerebrovascular diseases within a week after exposure. It is observed that the sequelae of cerebrovascular disease and ischemic stroke were the most sensitive subtypes to NO_2_ pollution, whereas hemorrhagic stroke did not exhibit a substantial association. Additionally, we also observed that the effects were stronger in the warm season, in patients with lower education and in those with single marital status. These findings provided new insights into the relationship between NO_2_ exposure and cerebrovascular mortality, and may have significant implications for the development of environmental and social policies.

## Figures and Tables

**Figure 1 toxics-12-00010-f001:**
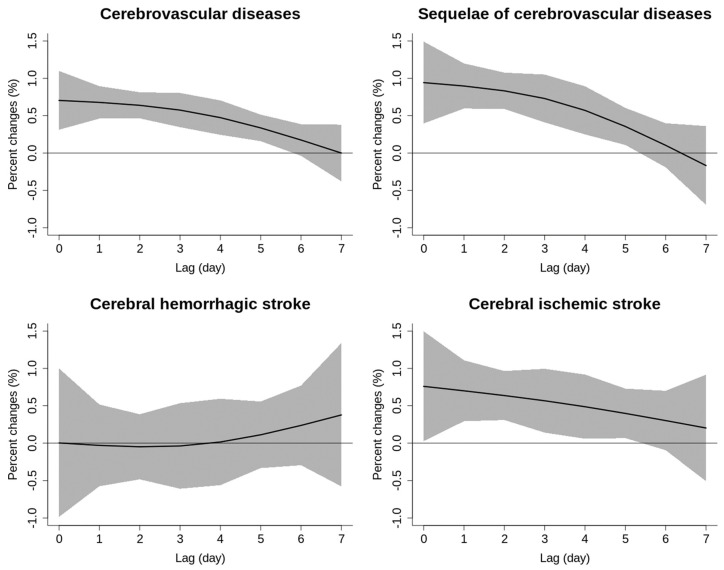
The lag structure for the percent changes of cerebrovascular mortality associated with a 10 μg/m^3^ increase in NO_2_.

**Figure 2 toxics-12-00010-f002:**
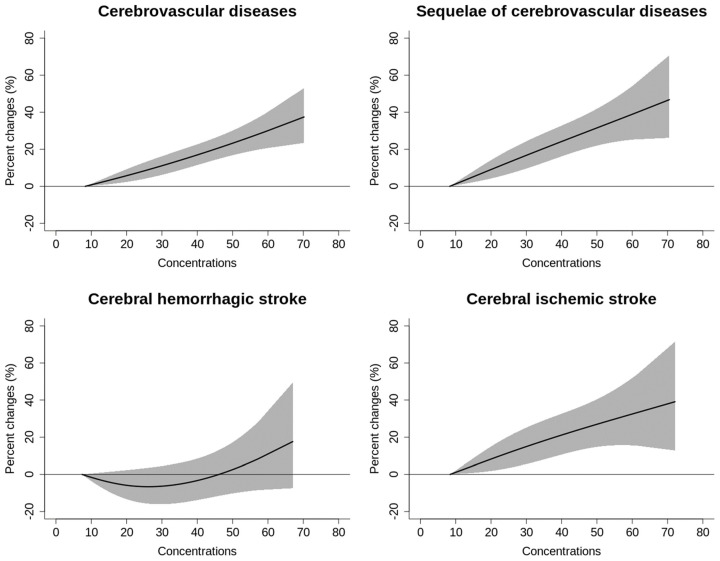
The cumulative exposure−response curves for the relationships between daily concentrations (μg/m^3^) of NO_2_ and cerebrovascular mortality over 0 to 7 days.

**Table 1 toxics-12-00010-t001:** Basic demographic characteristics of deaths for cerebrovascular diseases in Shanghai, from 2013 to 2021.

Variable	Cases (*n*, %)
Cerebrovascular deaths	219,147 (100.0)
Sequelae of cerebrovascular disease	111,182 (50.7)
Hemorrhagic stroke	37,539 (17.1)
Ischemic stroke	60,217 (27.5)
Other types of cerebrovascular disease	10,209 (4.7)
Sex	
Male	108,916 (49.7)
Female	110,231 (50.3)
Age (years)	
<80	67,437 (30.8)
≥80	151,710 (69.2)
Season	
Warm	96,437 (44.0)
Cold	122,710 (56.0)
Education	
≤9 years	85,529 (39.0)
>9 years	91,230 (41.6)
Unknown	42,388 (19.4)
Marriage	
Single	102,382 (46.7)
Married	116,460 (53.1)
Unknown	305 (0.2)

**Table 2 toxics-12-00010-t002:** Summary statistics for air pollution concentrations and meteorological variables in Shanghai, from 2013 to 2021.

Variables	Mean	SD	Min	P_25_	P_50_	P_75_	Max	IQR
NO_2_ (μg/m^3^)	23.8	12.5	1.6	15.7	20.9	28.4	133.7	12.7
PM_2.5_ (μg/m^3^)	43.1	29.4	1.0	22.7	35.2	54.6	440.3	31.9
PM_2.5–10_ (μg/m^3^)	27.2	18.7	0.0	16.0	23.3	34.6	442.1	18.6
O_3_ (μg/m^3^)	83.3	36.6	0.1	56.2	77.9	104.9	276.9	48.7
Temperature (°C)	15.9	8.5	−7.1	8.5	15.9	23.1	34.8	14.6
RH (%)	72.9	14.6	21.1	64.6	75.4	83.8	100.0	19.2

Abbreviations: PM_2.5_, particulate matter with an aerodynamic diameter ≤ 2.5 μm; PM_2.5–10_, particulate matter with an aerodynamic diameter > 2.5 μm and ≤10 μm; NO_2_, nitrogen dioxide; O_3_, ozone (8-h mean); RH, relative humidity; SD, standard deviation; Min, minimum; P_25_, 25th percentile; P_50_, 50th percentile; P_75_, 75th percentile; Max, maximum; IQR, interquartile range.

**Table 3 toxics-12-00010-t003:** Spearman correlation coefficients among the exposure variables including air pollutions and meteorological parameters in Shanghai, from 2013 to 2021.

	PM_2.5_	PM_2.5–10_	O_3_	Temperature	RH
NO_2_	0.54 *	0.32 *	−0.18 *	−0.38 *	−0.30 *
PM_2.5_	-	0.65 *	−0.04 *	−0.37 *	−0.31 *
PM_2.5–10_	-	-	−0.12 *	−0.14 *	−0.41 *
O_3_	-	-	-	0.52 *	−0.20 *
Temperature	-	-	-	-	0.37 *

* Statistically significant.

**Table 4 toxics-12-00010-t004:** Cumulative changes (mean and 95% confidence intervals) in cerebrovascular mortality associated with a 10 μg/m^3^ increase in NO_2_ over lag 0 to 7 days.

Categories	Cerebrovascular Diseases	Sequelae of Cerebrovascular Disease	Hemorrhagic Stroke	Ischemic Stroke
Total	3.62 (2.56, 4.69)	4.29 (2.81, 5.80)	0.54 (−2.01, 3.15)	4.30 (2.30, 6.33)
Sex				
Male	2.75 (1.26, 4.25)	2.80 (0.70, 4.94)	1.36 (−1.98, 4.82)	3.64 (0.77, 6.58)
Female	4.49 (2.99, 6.02)	5.72 (3.63, 7.84)	−0.66 (−4.54, 3.37)	4.91 (2.14, 7.75)
*p*	0.11	0.06	0.45	0.54
Age				
<80	3.32 (1.39, 5.28)	3.32 (1.39, 5.28)	−0.30 (−3.74, 3.27)	1.71 (−2.00, 5.57)
≥80	3.75 (2.49, 5.04)	4.84 (2.73, 6.95)	1.49 (−2.24, 5.37)	5.27 (2.92, 7.68)
*p*	0.71	0.49	0.50	0.12
Season				
Warm	5.35 (3.13, 7.62)	8.95 (5.75, 12.24)	−1.86 (−6.81, 3.34)	4.50 (2.20, 6.86)
Cold	2.91 (1.70, 4.14)	2.76 (1.08, 4.47)	1.06 (−1.91, 4.12)	2.82 (−1.23, 7.04)
*p*	0.06	<0.001 *	0.33	0.49
Education				
Low	4.68 (2.93, 6.47)	5.45 (3.08, 7.87)	−0.89 (−5.56, 4.01)	5.46 (2.13, 8.91)
High	2.05 (0.48, 3.64)	2.67 (0.35, 5.05)	0.89 (−2.43, 4.33)	2.15 (−0.77, 5.15)
*p*	0.03 *	0.10	0.55	0.15
Marriage				
Single	5.14 (3.58, 6.73)	5.07 (3.00, 7.18)	2.17 (−2.33, 6.87)	6.57 (3.60, 9.62)
Married	2.33 (0.89, 3.79)	3.53 (1.41, 5.69)	−0.19 (−3.26, 2.97)	2.40 (−0.28, 5.17)
*p*	0.01 *	0.31	0.40	0.04 *

* Statistically significant.

**Table 5 toxics-12-00010-t005:** Cumulative changes (mean and 95% confidence intervals) in cerebrovascular mortality associated with a 10 μg/m^3^ increase in NO_2_ over lag 0 to 7 days after adjusting for co-pollutants.

	Cerebrovascular Diseases	Sequelae of Cerebrovascular Diseases	Hemorrhagic Stroke	Ischemic Stroke
No adjustment	3.62 (2.56, 4.69)	4.29 (2.81, 5.80)	0.54 (−2.01, 3.15)	4.30 (2.30, 6.33)
+ PM_2.5_	2.95 (1.79, 4.13)	3.86 (2.23, 5.52)	0.63 (−2.19, 3.54)	2.77 (0.60, 5.00)
+ PM_2.5–10_	2.89 (1.78, 4.02)	3.19 (1.64, 4.76)	0.99 (−1.71, 3.77)	3.55 (1.45, 5.68)
+ O_3_	3.29 (2.23, 4.36)	3.93 (2.44, 5.44)	0.49 (−2.07, 3.11)	3.87 (1.87, 5.91)
+ all	2.81 (1.64, 3.99)	3.53 (1.90, 5.20)	0.81 (−2.03, 3.74)	2.73 (0.54, 4.97)

## Data Availability

The datasets used or analyzed during the current study are available from the corresponding author on reasonable request.

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
