# Peer review of "Residential Nitrogen Dioxide Exposure and Cause-Specific Cerebrovascular Mortality: An Individual-Level, Case-Crossover Study"

_toxics, 2023, doi:10.3390/toxics12010010_

Round 1
Reviewer 1 Report
Comments and Suggestions for Authors
please see my word document!

Comments on the Quality of English LanguageI suggested some minor changes (see my document)
Author Response
- I do not agree with the authors that a time-series study is of ecological design (line 59), while a case-crossover study uses individual data. Indeed, with casecrossover, you could also add individual data which would be more difficult in a time-series. But, as far as the current paper goes, as individual factors it only includes two broad age groups, marital status, educational achievements, and gender. These broad groups could also be analyzed in a timeseries in separate stratified analyses. I agree that checking for interactions and thus establishing the significance of differences between groups would be more of a challenge in a time-series design.
Reply: Thanks for this suggestion. We have deleted the sentence “However, most of them adopted an ecological time-series study design, which makes causal inference difficult” and rephrased this paragraph (see lines 60–62).
- I note that the authors claim that in their study they use individual exposure estimates. But I am not convinced that these are much better than estimates based on fixed monitors. NO2 displays a strong spatial variability. A 1 kilometer grid is by far not sufficient to capture this variability. Also, even in the 3 rd dimension, variability is strong. For example, in a street canyon, even the floor matters a lot when it comes to NO2 exposure. Therefore, I doubt that satellite based data are sufficient to estimate ground levels accurately. But in the end, it does not matter so much if you capture spatial differences correctly or if you even estimate absolute values correctly. In temporal comparisons, only the day-today difference is relevant. And as data from fixed monitors indicate, across a city NO2 levels move up and down roughly at the same amount, no matter how high the absolute values at a given location: monitoring data are highly correlated with each other over time. And even if data on a 1 kilometer grid were (slightly) better in capturing day-to-day differences than city-wide monitoring data, you still would not know how much time each person spent at his or her home address during the days of interest. Considering this, I am not so convinced that their claim in the discussion (“we obtained residential NO2 concentration with a higher resolution exposure model, allowing for a more precise estimation”, line 233) is entirely justified. This was just my comment to the first part of the paragraph starting at line 55.
Reply: Thanks for pointing this issue. Shanghai, as a megacity spanning 6,340 square kilometers, encompasses urban, suburban, and rural areas. Generally, a few monitoring stations distribute in urban areas, but cannot cover suburban and rural areas. Compared to monitoring stations, there is no doubt that exposure models can predict NO2 concentrations better in suburban and rural areas. Because there is little monitoring station in these areas, and monitoring data from urban areas is unsuitable as a substitute (see lines 270–274). We agree with the reviewer’s opinion that NO2 has a strong spatial variability. One of advantages of satellite data is that it can capture the spatial variability of NO2 better than the city-wide monitoring data. The high-precision exposure model used in our study has high accuracy of prediction, and the overall Cross-validation (CV) R2 and root mean square error (RMSE) value is 0.80 (7.78 μg/m3) compared to 1546 monitoring stations [1–3].
In addition, we do not know exactly the time each person spent at his or her home address. However, nearly 70% of the participants in our study were aged over 80, with an average age for the deceased people of 81.9. Therefore, we believe that these people are more likely to spend their time near their home address, since most of them have retired and may be physically challenged. We have rephrased the expression in discussion and limitation part of this study (see lines 348–355).
Reference
- Meng X, Liu C, Zhang L, Wang W, Stowell J, Kan H, Liu Y: Estimating PM2.5 concentrations in Northeastern China with full spatiotemporal coverage, 2005–2016. Remote Sensing of Environment2021, 253.
- Shi S, Wang W, Li X, Hang Y, Lei J, Kan H, Meng X: Optimizing modeling windows to better capture the long-term variation of PM2.5 concentrations in China during 2005–2019.Science of The Total Environment2023, 854.
- Li X, Wang P, Wang W, Zhang H, Shi S, Xue T, Lin J, Zhang Y, Liu M, Chen R et al: Mortality burden due to ambient nitrogen dioxide pollution in China: Application of high-resolution models. Environ Int2023, 176: 107967.
- That paragraph continues: “In addition, evidences of association between NO2 exposure and cerebrovascular disease were inconsistent.” (I would rather write: “evidence is inconsistent”!) But if the evidence is indeed inconsistent, we would expect this paper to alleviate that problem. In that case it is not sufficient to find another effect estimate that is consistent with some but not with other studies. At least the authors should discuss possible sources of inconsistence. Here, it seems, that they claim inconsistence in the introduction with the aim of motivating their study. In the discussion, they argue the same inconsistency away and point out (correctly!), that different effect estimates are mostly due to different lags (or single lags versus multiple lags) assessed. But of course, the issue of inconsistencies with NO2 effects goes deeper! Close to incineration sources, many pollutants are found in higher concentration, and ultrafine particles might be the most relevant. Because of poor data availability regarding ultrafine particles, their effect is severely understudied. Particle mass is only a very poor proxy of ultrafine particles and particle number. Depending on the sources and the local geography and meteorology, NO2 can serve a very good indicator of ultrafine particle exposure. So, some of the effects attributed to NO2, might indeed be due to other incineration products. And inconsistencies between studies might come from differences in local meteorology and source composition.
Reply: We appreciate the reviewer for this suggestion. Consistent with the reviewer’s opinion, the claim of inconsistence in the part of introduction is intended to clarify the motivation of our study. Additionally, we totally agree with the reviewer on the point that inconsistencies between studies results might come from differences in local meteorology and source composition. We have rephrased this paragraph and further discussed possible causes of inconsistence in the section of discussion. In addition to local weather and source composition of pollutants, we further discuss the impact of research design (time-series design versus case crossover design), exposure assessment methods (monitor data versus satellite model data), statistic model parameters (lag days and functions of model) and health data quality on effect of NO2 exposure on the risk of cerebrovascular mortality (see lines 63–64, lines 261–290).
- When analyzing effects on stroke mortality, the authors should somehow address the question: Does air pollution increase the risk of stroke or, given a stroke event, the risk of dying? I feel this is especially important for a toxicology journal, because toxicology is mainly about mechanisms, as opposed to epidemiology, which is about effect sizes and public health relevance in real world settings. At least by comparing different studies on morbidity and mortality, this aspect should be discussed in the paper.
Reply: Thanks for this suggestion. We have rephrased the sentences. Consistent with previous findings, we analyses the acute health effects of air NO2 pollution on cerebrovascular mortality, aiming to explore the risk of premature death caused by acute exacerbation of air NO2 pollution in cerebrovascular patients. The results of epidemiological research can provide important insights for toxicology research (see lines 227–230, lines 304–307).
- Inconsistencies regarding type of stroke (ischemic or hemorrhagic) could stem from poor health data. I assume that in some places, if a person dies on the street (or in the ambulance on the way to a hospital), and the symptoms clearly indicate a stroke, an autopsy is not always performed. And then, only based on the symptoms, a differentiation might be difficult. I’d be interested to learn how many diagnoses in the death certificates in Shanghai are based on autopsy!
Reply: We appreciate the reviewer for this very good point. We agree that cause of death is often determined by symptoms rather than autopsy, especially for mortality occurred outside of hospitals. We have added the contents in the methods section and rephrased this part in limitation paragraph. In real life, autopsy is rarely adopted for determining the cause of death of ordinary people, although it is a gold standard. Due to limited data acquisition, we are unclear which ones have been identified through autopsy, but it is believed that the proportion of autopsies is not related to air pollution, so it will not be a confounding factor. Death certificates in Shanghai are completed either by community doctors for deaths outside of hospital or by hospital doctors for deaths in hospitals. The information on the certificates was coded according to the International Classification of Diseases, Revision 10 (ICD 10). In shanghai, the cremation must be conducted with the death certificate, which reduces the proportion of missing death registration. In the current study, we collected 219,147 cases of cerebrovascular mortality from 1 January 2013 to 31 December 2021 from the mortality database of Shanghai municipal center for disease control and prevention. Among them, 65.28% diagnosed in hospitals and 33.46% diagnosed outside of hospital. For all the cases in hospitals, the cause of death was determined through a combination of symptoms and CT or MRI examinations, following standard medical norms, which is not difficult to distinguish the type of cerebrovascular diseases. For the cases outside the hospital, 99.55% diagnosis are made by community doctors based on medical records. And we believe that the percentage of misclassification would be very low in our study (see lines 92–97, lines 281–290, lines 345–348).
- I do struggle with the category of I69 as sequelae of cerebrovascular disease. This diagnoses describes a chronic state usually after repeated strokes. If people die, this might be due to another new stroke event, or due to some other causes. Mental and physical impairment might lead to traumatic events, paralysis might lead to poor lung ventilation and thus to respiratory infections. Therefore, as a cause of death, this diagnostic category is not very well defined. Maybe the authors would want to discuss their findings regarding NO2 effects on I69 in that light?
Reply: Thanks for pointing this important issue. We totally agree with the opinion of reviewer. Sequelae of cerebrovascular disease is a chronic state, which do not lead to death fundamentally. Its death is basically due to complications, including some infectious diseases and thrombotic diseases caused by paralysis, some traumatic events caused by mental and physical impairment, and some recurrent stroke events. We agree the reviewer’ opinion. We want discuss the effect of air NO2 pollution on population of cerebrovascular sequelae, although it’s not a well-defined diagnostic outcome, because the number of researches that focus on sequelae of cerebrovascular disease and air pollution is rare. We have rephrased the sentences in discussion section (see lines 281–290).
- Besides that, I only have some minor comments regarding grammar and wording: Line 95: “Other types of cerebrovascular diseases (I64-68) did not considered…”: “were not considered”! Line 129: “We conducted stratified analysis…”: “analyses”! Line 157: “but negatively correlations with O3 …”: “negative”! Table 3: the last line (RH) is not necessary and redundant. Line 221: “have been verified by numeral other studies.” Better: “a number of other studies” Discussion of differences in vulnerability (starting line 243): Having a higher risk of stroke or a higher risk of dying from stroke (e.g. in the case of living alone) is not the same as having a higher relative risk with increasing air pollution.
Reply: We have updated our manuscript as recommended and checked for language throughout entire text.
- More specifically, the way the differences between the seasons is discussed, is very sloppy. The study did not analyze hot days, but compared warm to cold season. Therefore, the argument that “high temperatures can lead to dysfunction of the vascular endothelium, increased platelet aggregation, cholesterol levels, and blood viscosity, thereby increasing the incidence rate of ischemic stroke” is not really to the point. Indeed, in the cold season, more deaths due to stroke were observed (56%) than in the warm season.
Reply: We really appreciate the constructive suggestions made by the reviewer. We realized that the argument mentioned is inaccurate and have deleted this sentence in the revised manuscript. We rephrased the sentence about the differences between the seasons (see lines 313–315).
- I am also not sure why more outdoor activities would lead to less measurement error regarding exposure. Will they engage in outdoor activities only in their 1-kilometer grid zone? And will it be people with an I69 diagnosis that likely engage in outdoor activities?
Reply: We appreciate this reviewer these issues. In previous epidemiological studies on air pollution, many have found the warm season effect, and the common explanations are that outdoor activities and opening windows for ventilation are more frequent in the warm season than in the cold season. This reduces the exposure error generated by the difference between indoor exposure and outdoor exposure when using outdoor monitoring or predicted exposure as a substitute for individual exposure (see lines 313–315) [1–2].
In our study, nearly 70% of participants were aged over 80, with an average age for the deceased people of 81.9. We believe that these people are more likely to spend their time near their home address, since a super majority of them have retired and may be physically challenged. Therefore, the 1-kilometer exposure model can capture environmental NO2 exposure in most cases (see lines 352–355).
Despite their limited physical activity and significantly less frequency of outdoor activities, patients with cerebrovascular sequelae are still more likely to engage in mild outdoor activities or opening windows for ventilation in warm seasons than in cold seasons, resulting in lower exposure errors in warm seasons.
Reference
- Huang F, Luo Y, Tan P, Xu Q, Tao L, Guo J, Zhang F, Xie X, Guo X: Gaseous Air Pollution and the Risk for Stroke Admissions: A Case-Crossover Study in Beijing, China. Int J Environ Res Public Health2017, 14(2):189.
- Zhang R, Jiang Y, Zhang G, Yu M, Wang Y, Liu G: Association between short-term exposure to ambient air pollution and hospital admissions for transient ischemic attacks in Beijing, China. Environmental science and pollution research international2021, 28(6), 6877–6885.
- Line 265: “Research have found” either: “Researchers have found” or: “Research has found” Line 268: “A double-blind randomized crossover study found that in individuals without existing health conditions, exposure to diesel exhaust has been found…”: there is simply one “found” to much
Reply: Thanks for the suggestion. We have updated our manuscript as recommended and check for other areas (see line 326, line 329).

Reviewer 2 Report
Comments and Suggestions for Authors
Review comments on paper 2758299: “Residential nitrogen dioxide exposure…”, Qian et al.
General Comments
In general, I found this paper very interesting and a delight to read. I recommend acceptance after the following comments are accommodated, including a few additional model runs that might be presented as a Supplement.
Recommended Changes
As in all time-series analyses, lag structures are important. I was delighted to see risks accumulated over lags 0-7, which few other studies have presented. However, Figure 1 raises questions about risks at even longer lags: are they positive, negative, or nil? If negative, the question of “harvesting” arises, that deaths during lags 0-7 were only advanced by a few days. In any event, the cumulative short-term risks should be compared to any available long-term risk estimates, of which they are a part. Few studies have made this comparison that is fundamental to causality: Short-term effects follow pre-existing ailments; long-term effects are presumed to have initiated new ailments.
Indoor air quality must be considered in more detail, especially given the attention being given to gas cookstoves in the United States. Presenting mean pollution levels by subgroup (season, gender, education) might indicate possible thresholds in dose-response due to neglecting indoor exposures.
The computer-generated air quality data should be compared to measured data for each pollutant and over time.
Multiple pollutant risk estimates should be presented, especially NO2 jointly with particulates, which are being emphasized worldwide. I would very much like to see risk estimates for NO2 , fine, and coarse particulates in the same regression.
Author Response
Reviewer 2:
- As in all time-series analyses, lag structures are important. I was delighted to see risks accumulated over lags 0-7, which few other studies have presented. However, Figure 1 raises questions about risks at even longer lags: are they positive, negative, or nil? If negative, the question of “harvesting” arises, that deaths during lags 0-7 were only advanced by a few days. In any event, the cumulative short-term risks should be compared to any available long-term risk estimates, of which they are a part. Few studies have made this comparison that is fundamental to causality: Short-term effects follow pre-existing ailments; long-term effects are presumed to have initiated new ailments.
Reply: Thanks for this suggestion. Previous studies on the acute effects of air pollution usually focused on the effects within a week of lag. Therefore, we chose 1 week as the maximum lag a priori, and found that the effect just disappeared after a lag of about one week. Therefore, the cumulative effect was calculated using lag0-7. According to the reviewer’s suggestion, we explore the effects of longer lag days with a maximum lag of 14 days and discover that the significant effects disappear within a week, while there is no significant effect on the second week (see Figure S1 in supplemental material). In addition, our study adopted the case-crossover design, which is a study design to explore short-time effects and unable to compare with long-term effects fundamentally. We sincerely wish that cohort studies can be conducted to explore related issues in the future.
- Indoor air quality must be considered in more detail, especially given the attention being given to gas cookstoves in the United States. Presenting mean pollution levels by subgroup (season, gender, education) might indicate possible thresholds in dose-response due to neglecting indoor exposures.
Reply: Thanks for this suggestion. Our study adopted the case-crossover design, which is a study design that focuses on the health effects of daily fluctuations in outdoor air pollution. However, indoor air pollution is usually relatively stable on a daily scale, such as that caused by cookstove, so it should not be a significant confounding factor and may be an effect modifying factor. However, due to limited data acquisition, we cannot explore this modifying effect, which has been listed as a limitation (see lines 343–345). Due the study design and relative stable indoor pollution over a short time, it also should not to be a significant confounding factor, therefore, the threshold of the exposure-response relationship curves is not necessarily related to indoor air pollution (see Figure S2 in supplemental material).
- The computer-generated air quality data should be compared to measured data for each pollutant and over time.
Reply: Thanks for this suggestion. Please refer to the references. The high-precision exposure model used in our study has high accuracy and high consistence with measured data, and the overall Cross-validation (CV) R2 range from 0.80 to 0.84 compared to 1546 monitoring stations [1–3] (see lines 105–109).
Reference
- Meng X, Liu C, Zhang L, Wang W, Stowell J, Kan H, Liu Y: Estimating PM2.5 concentrations in Northeastern China with full spatiotemporal coverage, 2005–2016. Remote Sensing of Environment2021, 253.
- Shi S, Wang W, Li X, Hang Y, Lei J, Kan H, Meng X: Optimizing modeling windows to better capture the long-term variation of PM2.5 concentrations in China during 2005–2019.Science of The Total Environment2023, 854.
- Li X, Wang P, Wang W, Zhang H, Shi S, Xue T, Lin J, Zhang Y, Liu M, Chen R et al: Mortality burden due to ambient nitrogen dioxide pollution in China: Application of high-resolution models. Environ Int2023, 176: 107967.
- Multiple pollutant risk estimates should be presented, especially NO2 jointly with particulates, which are being emphasized worldwide. I would very much like to see risk estimates for NO2, fine, and coarse particulates in the same regression.
Reply: Thanks for this suggestion. We have added the multiple pollutant analysis in methods section (see lines 143–146). In sensitive analysis, we conducted the two-pollutant analysis adjusting of PM2.5, PM2.5-10 and O3, respectively; and further performed multiple pollutant risk estimates adjusting of PM2.5, PM2.5-10 and O3, with the result that the effects of NO2 on cause-specific cerebrovascular mortality are robust (see Table 5). Considering the limited length of an article, it is difficult to comprehensively present and discuss the results of all pollutants.

Round 2
Reviewer 2 Report
Comments and Suggestions for Authors
I am satisfied with the changes and recommend acceptance.